# Understanding and Involving the Perspective of Pregnant Women as Users When Designing the Framework of e-Health and Exercise Interventions during Pregnancy: Preliminary Study

**DOI:** 10.3390/healthcare12111121

**Published:** 2024-05-30

**Authors:** Rita Santos-Rocha, Mariana Ferreira, Nuno Pimenta, Marco Branco, Miguel Oviedo-Caro, Anna Szumilewicz

**Affiliations:** 1ESDRM—Department of Physical Activity and Health, Sport Sciences School of Rio Maior, Santarém Polytechnic University, 2040-413 Rio Maior, Portugal; marianaferreira_24@hotmail.com (M.F.); npimenta@esdrm.ipsantarem.pt (N.P.); marcobranco@esdrm.ipsantarem.pt (M.B.); 2SPRINT—Sport Physical Activity and Health Research and Innovation Center, 2040-413 Rio Maior, Portugal; 3CIPER—Interdisciplinary Centre for the Study of Human Performance, Faculty of Human Kinetics (FMH), University of Lisbon, 1499-002 Cruz Quebrada, Portugal; 4Department of Physical Education and Sport, Faculty of Education Sciences, University of Seville, 41080 Seville, Spain; moviedo@us.es; 5Faculty of Physical Education, Gdansk University of Physical Education and Sport, 80-336 Gdansk, Poland; anna.szumilewicz@awf.gda.pl

**Keywords:** pregnancy, postpartum, physical activity, exercise, exercise testing, exercise prescription, health, mobile app, e-health, m-health

## Abstract

Health and exercise technology may promote a healthy lifestyle during pregnancy. The objective of this cross-sectional study was to understand and involve the perspective of pregnant women as users in the design of a framework for future e-health and exercise interventions during pregnancy. Pregnant women replied to a questionnaire aimed at understanding their physical activity patterns, needs, and preferences regarding the use of mobile applications (apps). The main results showed that one-third of the women did not practice any type of exercise during pregnancy. Women preferred to exercise in a gym, outdoors, or at home. The majority already had or were currently using a fitness app, but never used any pregnancy-specific app. Most women agreed that it was important to have a specific app for pregnancy to improve knowledge about recommendations on lifestyle, have direct contact with health and exercise professionals, have social interaction with other mothers, and have guidance on preparation for childbirth and postpartum recovery. Understanding and involving the perspective of pregnant women as users will allow researchers to improve the design of a pregnancy-specific app and future e-health and exercise interventions during pregnancy. These preliminary results will lead to the development of the “active pregnancy app” focused on the promotion of an active and healthy lifestyle during pregnancy and postpartum.

## 1. Introduction

Pregnancy is a unique physiological event during which various adaptations occur in the woman’s body to ensure the normal development and well-being of the fetus. Pregnancy is, therefore, a stage of physical, behavioral, and social changes for women [1]. This period is considered an opportunity for a more active and healthier lifestyle. Women have more frequent visits to the healthcare provider during pregnancy and postpartum, which can lead to a greater motivation to improve their health during this period. Healthy maternal behaviors such as decreasing caffeine consumption and smoking, increasing physical activity, and maintaining healthy weight gain have been shown to decrease the risk of pregnancy-related morbidities [2]. Physical activity and exercise have various health benefits for pregnant women [3]. Current scientific evidence supports that prenatal physical activity that is practiced according to the appropriate recommendations benefits both fetal and maternal physiological, psychological, and health conditions [4]. A recent scoping review by Hayman et al. [5] provides a summary of public health recommendations for physical activity during pregnancy around the world. However, implementing exercise into daily life and long-term adherence remains challenging. Even though there are multiple research results about exercise during pregnancy, women are still misinformed about this practice, and very few perform physical activity during this phase [6]. Moreover, the prevalence of physical activity during pregnancy is affected by the maternal age group and educational level [7]. Due to the low prevalence of physical activity in women of reproductive age and the high prevalence of chronic conditions in the prenatal phase, such as obesity, hypertension, and diabetes, it is essential to increase physical activity levels before, during, and after pregnancy [8]. Indeed, less than 15% of women actually fulfill the minimum recommendation of 150 min per week of moderate-intensity physical activity or 75 min of vigorous-intensity physical activity during pregnancy [8]. Women often face barriers to engaging in exercise, which include time constraints, lack of motivation, lack of social support, and lack of resources, among others [9]. Other barriers may include resource limitations, financial costs, and knowledge which inhibit the widespread implementation and dissemination of these beneficial programs and guidelines.

Digital health tools such as applications (apps) are being increasingly used by women to access pregnancy-related information [10]. Health and exercise technology (e-health) may address some physical activity and exercise barriers, promote health education, and deliver effective and scalable interventions during pregnancy. Incorporating tailored exercise programs into a technology-based intervention in the home and outdoor environments may help overcome these barriers and promote health benefits. New and developing e-health (electronic), m-health (mobile), and remote monitoring technologies offer a potentially transformative and cost-effective solution to increase access to physical activity and exercise programs for pregnant women. Yet, its effectiveness in changing lifestyle remains unknown [11]. During pregnancy, women increasingly turn to m-health for access to health information and support. This demand for the use of mobile health apps during pregnancy presents a unique opportunity for the development of apps at a time when women are generally more motivated to optimize their health and change their lifestyles. Smartphone-based interventions show considerable promise for increasing physical activity levels. In 2019, smartphones were the most used devices for access to the Internet (87%), followed by laptops (69%) and tablets (56%) [12]. Exercise apps during pregnancy bring with them numerous positive aspects; specifically, they can be adapted to the target population, can be performed anywhere and at any time, and are interactive and accessible to the majority of the population, regardless of socioeconomic level, since around 96% of women between 8 and 49 years of age have a smartphone [13]. Paradoxically, pregnancy is the medical condition with the largest number of apps available; however, mobile apps designed specifically to increase physical activity during pregnancy are scarce, and user interaction with these technologies is critical and culturally specific [14]. Moreover, available apps show low incorporation of behavior change techniques [5], and those that do focus on such an objective do not follow the current physical activity guidelines [15]. Another problem with exercise apps designed for pregnancy is that they do not consider current evidence-based physical activity guidelines, prior screening for contraindications to physical activity, adequate personalization resources to take into account the characteristics of the pregnant woman, and the involvement of qualified exercise professionals during app development [2]. Tinius [16] concluded that existing free mobile apps “are insufficient to enable women to achieve recommended levels of physical activity during pregnancy and postpartum”. Muñoz-Mancisidor et al. [17] concluded that only a small percentage of free pregnancy apps available (in Spanish) should be recommended.

Nevertheless, there are pregnancy and exercise apps based on evidence regarding the potential to address the lack of exercise among pregnant and postpartum women [18], to increase perceived benefits and enjoyment [19], and to provide social support for exercise [20]. Another example is the SmartMoms Canada app, developed by Adamo et al. [21] to promote adequate gestational weight gain and other healthy behaviors. However, even though this app was built and tested based on evidence [22,23], it is not available for free.

Given this lack of specific application based on scientific evidence for pregnancy, there is a clear need to create an application that responds to the needs mentioned above. Moreover, there are no such apps available in the Portuguese language. It is also crucial to utilize the perspectives identified by potential users [21,23].

The objectives of this cross-sectional study were to understand the physical activity patterns during pregnancy, to identify the needs and preferences of pregnant women regarding the use and development of a tailored e-exercise mobile app, and to involve the perspective of pregnant women as users in the design of a framework of future e-health and exercise interventions during pregnancy. 

## 2. Materials and Methods

### 2.1. Study Type

A cross-sectional study was carried out with pregnant or postpartum women by completing a questionnaire.

### 2.2. Participants

We invited pregnant or postpartum women to reply to a questionnaire to help us understand their physical activity patterns, needs, and preferences regarding the use of apps. The inclusion criteria were Portuguese women, aged 18–50 years, with an understanding of the Portuguese language, who were pregnant or in the postpartum period (up to one year postpartum) at the time they replied to the questionnaire.

### 2.3. Ethical Considerations

This study received approval from the Ethics Committee of the Santarém Polytechnic University (nr. 6/2021 ESDRM). All procedures applied to participants were in accordance with the ethical standards of the institutional research committee and the 1964 Helsinki Declaration. All participants were informed about the objectives and nature of this study, as well as the benefits and details of their participation. They were also informed of their right to withdraw from this study at any time. The confidentiality of their identities was also guaranteed. To consent, the process was conducted as follows: after reading the following statement, “I declare that I have understood the written information provided to me regarding the objectives of the study for those responsible, as well as the guarantee of the possibility of, at any level, refusing to participate without any consequences. In this way, I give my consent and agree to participate in this study”, they confirmed their participation in the study and selected the “Next” option.

### 2.4. Questionnaire

The digital questionnaire entitled “APP–Gravidez Ativa: promoção do estilo de vida ativo e saudável durante a gravidez e no pós-parto” (Active Pregnancy App–Active and healthy lifestyle promotion during pregnancy and postpartum; Portuguese) includes 33 questions (Appendix B) and was designed to be answered quickly and intuitively, facilitating the response process (Appendix A).

### 2.5. Data Collection and Analysis Procedures

Questionnaire implementation lasted 6 months, taking place between June and November 2022. A convenience sample was used. The participants independently responded to the questionnaire, which was in an online format and was shared with several groups of pregnant women on Facebook, as well as spread “word of mouth” in order to reach the largest number of women possible.

A descriptive analysis was performed regarding the frequency in percentage.

## 3. Results

### 3.1. Participants’ Characteristics

The study sample was composed of 105 Portuguese women, between 28 and 48 years of age. The response rate was 100%. Most women resided in the Lisbon and Vale do Tejo region (81%), while the remainder resided in Azores (2.9%), Alentejo (2.9%), Algarve (2.9%), Centro (3.8%), Madeira (1%), North (3.8%), and outside Portugal (3.8%). Regarding education level, 46.7% had a higher education level, 48.5% had a secondary education level, and only 4.8% had a basic education level. Finally, all the women who responded to the questionnaire were already mothers or were pregnant.

### 3.2. Physical Activity Practice during Pregnancy

Regarding physical activity (PA) practice during pregnancy, 36.2% of women did not practice any type of PA during pregnancy; 14.3% of women practiced PA 1 to 2 days a week; 37.1% of women practiced PA 3 to 4 days a week; 4.8% of women practiced PA 5 to 6 days a week; and 7.6% of women practiced PA every day.

Regarding the time in daily minutes that they dedicate to practicing PA, 36.2% did not carry out any type of activity (0 min), 23.8% reported a practice time of 30 min/day, 19% reported 45 min/day, and 21% reported 60 min/day.

The activity most performed by women at this stage was clearly walking; however, there were other activities that pregnant women preferred, such as swimming, water aerobics, Pilates, and yoga. Some women (i.e., 14.3%) also mentioned their participation in childbirth preparation classes.

### 3.3. Physical Activity Practice during the Postpartum Period

In the postpartum period, the percentage of mothers who did not practice any type of exercise was 33.3%, followed by practicing PA 1 to 2 days/week, which was the regularity preferred by women (43.8%); PA 3 to 4 days/week (21.9%); and a minority (1%) practicing all days during the postpartum period.

The time in minutes/day spent in practicing PA depends on the results of practicing PA during pregnancy, with a preference of 30 min/day, and a maximum time of 60 min/day.

Finally, regarding the type of PA during this period, the preference is also very similar to what was mentioned during the pregnancy period, with walking being the most practiced activity, followed by Pilates, water aerobics, exercises at home, yoga, and step, among other activities.

### 3.4. Practice of Physical Activity under the Supervision of an Exercise Professional

Regarding the practice of PA under the supervision of an exercise professional during pregnancy, 75.2% of women did not have any type of supervision; though in the postpartum period, this percentage was reduced, still 61.9% of women did not have any type of supervision.

### 3.5. Preferred Location for Physical Activity

Women preferred to practice PA in a gym (69.5%), outdoors (44.8%), or at home (39%). The practice of PA in the workplace (0.02%) or in sports associations (0.07%) is a minority preference.

### 3.6. Use of Computer App for Physical Activity

The majority of women surveyed (93.3%) have already used or are currently using a PA application (APP). The most well-known and used apps were Strava, MiFit, Nike Training Club, Leap Fitness, FitOn, and exercise videos on YouTube.

Regarding specific apps for pregnancy or postpartum, 89.5% of women did not use any specific app.

Regarding the regularity of app use by women, the majority used them between 3 and 4 days per week (40%) and 5 and 6 days per week (13.3%). It can be concluded that this is an effective way used by women to practice PA regularly at this stage of life, whether during pregnancy or postpartum.

### 3.7. Preferences Regarding the Future “Active Pregnancy App”

Finally, and to better understand the needs and preferences of pregnant and postpartum women, the questions were limited only to the future “Active Pregnancy APP”. Most women (98%) agreed that it was important to have a specific app for pregnancy and postpartum to improve knowledge about recommendations for an active lifestyle, recommendations for practicing PA, and recommendations for a healthy lifestyle, including diet, sleep, and health. Regarding PA sessions, PA was more preferred at home (48.6%) than outdoors (39%). Women unanimously agreed that it would be important in a pregnancy and postpartum app to have direct contact with both health professionals and exercise professionals, as well as guidance on preparation for childbirth (78.1% agreed and 9.5% totally agreed) and about recovery after childbirth (74.3% agreed and 10.5% totally agreed).

Regarding the recording of health parameters and recording the PA pattern or training plan, 77.1% agreed with its importance, and 20% completely agreed with its importance. There was also a consensus on the importance of counting daily steps in the app (81.9% agreed and 12.4% completely agreed) and diet plan records (70.5% agreed), making it a multidisciplinary app.

Finally, regarding interoperability with other apps and social networks, there was also consensus, with the majority (76%) of women agreeing with its importance.

Aiming that the future app will respond to the needs of pregnant women and mothers, an open question was asked: “Suggestions that you would like to make regarding the functionality and usefulness of a specific fitness app for pregnancy or for postpartum?”. Many of the women did not want to add any suggestions; however, there were those who made suggestions (25.7%), increasing the value of this questionnaire, with answers such as follows:PA and exercise guidance: “e*xistence of group classes in the APP”, “Zumba workouts”, “Pilates”, “APP with exercise videos and various options depending on mobility throughout the pregnancy and in postpartum recovery”, “training plans with exercise images”, “exercises to do with the baby”, “have exercises to do both in the gym and at home, in various contexts”, “exercise levels, for example: having the option of easy and difficult*”;Exercise monitoring: “*having connection with the smartwatch”*;Health education guidance: “*advice or ‘did you know’ periodically about what pregnant or postpartum women may feel when exercising and may or may not be normal (example: having cramps or leaking urine)*”;Motivation tools and alerts: *“having motivational and control alerts for various indicators”, “have notifications with reminders of healthy habits, such as drinking water, walking, etc.”, “notifications and motivational messages*”;Social interaction: “*APP with forums with other mothers”, “having access to a private group with more mothers”, “have a direct connection with Facebook and Instagram”, “have a ranking of the most active pregnant women*”;Professional interaction: “*having direct contact with an exercise professional*”;Technical features: “*that the APP uses little memory on the smartphone”, “free APP*”.

## 4. Discussion

The objective of this cross-sectional study was to understand and involve the perspective of pregnant women as users in the design of a framework for future e-health and exercise interventions during pregnancy.

The main results showed that 36.2% of women did not practice any type of exercise during pregnancy. Women preferred to practice exercise in a gym (69.5%), outdoors (44.8%), or at home (39%). The majority of women (93.3%) have already used or currently use a fitness app; however, 89.5% of women never used any pregnancy-specific app. Most women agreed that it is important to have a specific app for pregnancy to improve knowledge about recommendations for an active and healthy lifestyle, have direct contact with health and exercise professionals, have social interaction with other mothers, and have guidance on preparation for childbirth and postpartum recovery.

According to Hughson et al. [24], pregnancy is the medical condition with the greatest number of applications available. However, mobile apps designed specifically to increase PA during pregnancy are scarce, and those that do exist do not consider current evidence-based guidelines [24], such as Canadian (2019) [25], American (2020) [26], Brazilian (2021) [27], Australian (2022) [28], and Polish (2023) [29] guidelines.

The results of this study were in line with this fact because 93.3% of women have or have had a fitness app, but only 10.5%, which is equivalent to 10 women, have or have had one app specifically for pregnancy and postpartum. This leads to an obvious conclusion: there is a lack of apps for this specific population, and those that exist are not captivating to the point of leading women to install the app. Another important result to point out is the place where they prefer to practice PA, leading to the conclusion that the recommended places for prescribing PA will be gyms, outdoors, and at home. The goal of the app is to make PA accessible with just one click. Women who practiced PA during pregnancy tend to reduce the time or eliminate the practice, becoming an important point to remember since it is essential at this stage that the woman has the ability to also focus on herself, her health, and her well-being. Therefore, in line with Hayman et al. [13], future app development should identify and adopt factors that enhance and encourage user engagement.

### 4.1. Strengths and Limitations of the Study

This study highlights the gaps and needs of pregnant and postpartum women and should inform all stakeholders designing pregnancy digital healthcare. As far as we know, this is the first study addressing Portuguese women’s preferences regarding the use and features of a specific app aimed at promoting tailored exercise during pregnancy and postpartum. In this way, with this questionnaire and its open question, women’s preferences regarding a specific pregnancy and postpartum app are clarified, enriching the creation of the app in terms of possible content.

The main limitation of this study was the reduced number of respondents (N = 105). On the other hand, the results may not reflect Portuguese society, given that the majority of women were living in the urban region of Lisbon.

### 4.2. Implications for Future Research

The currently existing scientific evidence supports the importance of physical exercise during pregnancy and the postpartum period. Due to the specificity of this population, research is necessary on the best adherence strategies, with specific objectives, to promote greater adaptation and greater safety, responding to the needs of pregnant women. The majority of pregnant women do not practice PA according to current guidelines, and one of the solutions could be the creation of an app that responds to their needs, given that the majority of women have a smartphone but most do not have a specific app for pregnancy and postpartum exercise.

It is essential to consider the opinions of women and their suggestions, and it has been suggested several times to include on the app groups/forums among more and future mothers; that the app does not consume much of the smartphone’s memory; that it has signs and notifications with the aim of motivating; and finally, it not only has training plans, but also group classes, such as Pilates and Zumba.

Based on this study, it will be possible to build a more structured app, based on current guidelines, making the intervention more appropriate and safer for women. Although one of the limitations was the small size of the sample, the women’s opinions regarding an app of these characteristics and their high interest in creating it, as well as their preferences, were clear. For future studies, it will be essential to increase the sample, and that the sample does not reside mainly in a specific place in the country but in several locations, so that it is as representative as possible. Surveys conducted in other countries would also be valuable in the context of creating an app for the international market.

### 4.3. Implications for Professional Practice

The development of a specific app will provide exercise and health professionals with an evidence-, professional expertise-, and consumer-based needs and preferences tool aimed at supporting the promotion and implementation of effective programs. In the future, such an app could be recommended by health and exercise professionals.

## 5. Conclusions

The lack of free, multifunctional, and evidence-based specific apps for healthy lifestyle, physical activity, and exercise guidance during pregnancy and postpartum is a fact. Understanding and involving the perspective of pregnant women as users will allow researchers to improve the design of a pregnancy-specific app and future e-health and exercise interventions during pregnancy. Moreover, such an app should be incorporated into m-health systems in the future.

This preliminary study allowed us to understand the needs and preferences of a population of pregnant and postpartum women in order to effectively develop the contents and features of the future Active Pregnancy App.

## 6. Patents

The future “active pregnancy app” will be a patent to be registered resulting from the work reported in this manuscript.

## Data Availability

Data will be made available to other researchers upon request. Original data in the Portuguese language can be found at Santarém Polytechnic University repository: https://repositorio.ipsantarem.pt/handle/10400.15/4388 (accessed on 4 April 2024).

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
