# Peer review of "Understanding and Involving the Perspective of Pregnant Women as Users When Designing the Framework of e-Health and Exercise Interventions during Pregnancy: Preliminary Study"

_healthcare, 2024, doi:10.3390/healthcare12111121_

Round 1
Reviewer 1 Report
Comments and Suggestions for Authors
Title: the title is long but succinct for the topic of the manuscript.
Abstract:
1. This sentence is missing a word “Understanding and involving the perspective of women as consumers will allow [researchers?] to improve the design of a pregnancy specific app and future e-health and exercise 28 interventions during pregnancy.
Introduction:
1. Line 40 please add “the” in front of “healthcare provider.”
2. Line 42, it sounds better to state “decrease risk of pregnancy-related” than improve the risk. Please update.
3. Line 43-44: this list is not a list of morbidities. Please update the sentence. It reads like the authors are trying to state that healthy maternal behaviors are related to other healthy behaviors, such as decreased caffeine consumption and smoking, increased PA.
4. Please update the fetus’ and the woman’s to fetal and maternal physiological…
5. Line 54 should the “on” be “in”?
6. The sentence that starts on Line 54 “due to the low prevalence …” goes until Line 59. This is too long. Please break into 2 sentence for clarity.
7. Should there be “which” before inhibit on line 62?
8. Line 75 the “at time” should read “at a time”
9. Line 85-86: Please clarify the statement “user (delete s’) interaction with these technologies is critical and culturally specific…” In general, the sentence from line 84 to 87 is too long and it is not clear. Please update for clarity.
10. The last sentence Munoz-Mancisidor et al seems out of place, since this does not seem to be focused on Spanish.
11. There are pregnancy and exercise apps based on evidence.
a. https://www.sciencedirect.com/science/article/pii/S2211335523003765
b. https://digitalcommons.wku.edu/ijes/vol14/iss7/5/
c. BumptUp: https://www.mdpi.com/2071-1050/14/19/12801
d. https://www.dovepress.com/obstetric-patients-and-healthcare-providers-perspectives-to-inform-mob-peer-reviewed-fulltext-article-IJWH
e. https://pubmed.ncbi.nlm.nih.gov/34055180/
f. http://www.adamolab.com/smartmomscanada-appfeatures.html
g. https://www.ncbi.nlm.nih.gov/pmc/articles/PMC8811692/
h. https://www.ualberta.ca/kinesiology-sport-recreation/news/2022/may/pregnancy-apps.html
i. https://www.researchgate.net/publication/348536138_Mobile-application_Intervention_on_Physical_Activity_of_Pregnant_Women_in_Iran_During_the_COVID-19_Epidemic_in_2020
12. Based on the topic, the introduction should review the current state of what IS available regarding exercise and pregnancy mobile apps: those based and not based on evidence. This is a critical review that is missing and specifically relates to this topic. Please update in order to understand how this fits in with the current state of literature.
Methods:
1. Please clarify the lines 120 to 125. The sentence starting on line 120 seems incomplete. Further, the sentence starting on line 124 does not make sense; please clarify.
2. Not sure what is meant by “mouth in mouth;” is this referring to the “word of mouth” expression?
Results:
1. Line 141 states the women were up to 48 years old, and the inclusion was either pregnant or pregnant within a year. Therefore, please confirm that this is the case for all participants.
2. The sentence starting on line 153 with “Regarding the time in daily minutes …” does not seem to have all of the percentages for each group (i.e. 30 – 60 min/day). Please update the sentence or put data in a table.
3. Line 159 mentions “childbirth preparation classes;” however, these are not typically done regularly. Please clarify if the questionnaire and responses are for regular PA and what that is defined as specifically.
4. Section 3.5 Line 179, please clarify the percentages that practiced PA in the workplace or sports association.
5. Line 186-187, please clarify the percentages that used the APPs 3-4 days/wk vs. 5-6 days/wk.
Discussion:
1. Line 252: although this is a Portuguese population and for the data to be more generalizable, the word “Portuguese” can be deleted for this sentence on line 252.
2. Line 259 probably the “in your” can be removed from before well-being to state simply “her health and well-being.”
3. Strengths and limitations line 263, it state the gaps and needs of Portuguese women; however the title and intro do not specifically focus on Portuguese women. Please clarify if this is a study using Portuguese women to apply to all pregnant and postpartum women OR if it is a study to focus on only Portuguese pregnant and postpartum women.
Conclusion:
1. Line 303, it seems there is a word missing in the sentence “Understanding and involving the perspective of women as consumers will allow [researchers?] to improve …”
2. Line 307, again, it seems there is a word missing “This preliminary study allowed [us?] to understand the needs and …”
3. Do you want the summery and article to focus on only Portuguese women as presented in the conclusion and discussion, then the title, abstract, and intro need to be updated related to this focus. Otherwise, just state pregnant and postpartum women.
Table & Figures: NA
Overall:
This is a timely topic, however, the introduction fails to detail the current state of literature related to evidence-based exercise/PA apps for pregnant and postpartum women. It is unclear if this is meant to be an article focused on Portuguese women or for a broader audience.
Comments on the Quality of English Languagethis is fine, there are only minor grammar changes.
Author Response
Thank you very much for your help in improving the paper.
Please see the attachment.

Reviewer 2 Report
Comments and Suggestions for Authors
Good job, but it could have been better. Please consider the below:
Introduction
Abbreviations should be written in full the 1st time with the abbreviations in brackets e.g. for apps and m-health.
Lines 64 & 73: “Digital health tools such as apps are being increasingly used by women to access pregnancy-related information.... During pregnancy, women increasingly turn to m-health to access to health information and support.” Any figures to support these statements?
Line 82: “…since around 96% of women 82 between 8 and 49 years old have a smartphone.” Is this value for Portugal? If no, are there any similar values for Portugal?
Materials and Methods
Lines 107 & 110: Postpartum is usually 6 weeks post delivery but you mentioned “postpartum” and “pregnant until one year before” as participants, which is confusing. Which one is it?
How did you arrive at your choice of using Facebook? What’s your exclusion criteria? What did you mean by “several groups of pregnant women on Facebook”? What is “mouth in mouth”?
What sampling method did you use?
How did you define physical activity?
Also, a provision of the following would have been helpful:
- Definition of terms
- Description of the design of the questionnaire
- The data collected
- Data analysis
Results
Line 141: What’s your denominator? And response rate?
Line 184: “Regarding specific APPs for pregnancy or postpartum, 89.5% of women did not use any specific APP.” Why? Because it did not exist, or they did not know it existed, or it existed but not in Portuguese language? We cannot just conclude they did not use a specific APP because it did not exist since it was not captured.
Line 192: How many women agreed that it was important to have a pregnancy specific app?
Line 202: What did you mean by agree vs completely agree?
Line 207: “Aiming that the future APP will respond to the needs of pregnant women and mothers, an open question was asked: “Suggestions that you would like to make regarding the functionality and usefulness of a specific fitness APP for pregnancy or for postpartum?”. Many of the women did not want to add any suggestions, however there were those who made suggestions…” So how many women made the suggestions?
Line 212: Providing a few quotes is good. But it would have been good to also flesh them out and provide some meaning instead of just giving us what they said verbatim. Like those are the quotes, so what?
Discussion
It would have been helpful to also capture the following:
- Reasons behind those results findings either in the questionnaire or in the discussion.
- Factors influencing doing a physical activity in these women
- Barriers to using the app and how to mitigate
- A pdf or Word document of the actual questionnaire instead of a google doc link that is not in English language.
Conclusion
Line 301: Perhaps I missed that information, but I did not see any result finding that confirmed the pregnancy specific app did not exist or that they would use it if it did.
Comments on the Quality of English LanguageNeeds editing. Seems English is not their 1st language
Author Response

(The authors gave the same response as above.)

Reviewer 3 Report
Comments and Suggestions for Authors
Overall Comments:
The authors present an observational study examining women's perspective as consumers in designing a framework for future e-health and exercise interventions during pregnancy.
The authors found that understanding and involving the perspective of women as consumers will allow to improve the design of a pregnancy-specific app and future e-health and exercise interventions during pregnancy.
I found the study carefully designed and developed, the question posed by the authors is well defined on page 2 (lines 64-69): “Digital health tools such as apps are being increasingly used by women to access pregnancy-related information. Health and exercise technology (e-health) may address some physical activity and exercise barriers, promote health education, and deliver effective and scalable interventions during pregnancy. Incorporating tailored exercise programs into a technology-based intervention in the home and outdoor environments may help overcome these barriers and promote health benefits.”
The methods are appropriate, the authors have done intensive data collection work.
The discussion and conclusions are well-balanced and adequately supported by the data.
The study is rigorous and of high scientific quality.
Minor Comments:
Please consider changing CONSUMERS to PREGNANT WOMEN USERS.
Please clarify in the abstract that this is a cross-sectional study, that is important for readers.
Author Response

(The authors gave the same response as above.)

Round 2
Reviewer 2 Report
Comments and Suggestions for Authors
Thank you for your responses. Please see the below:
Materials and Methods
A provision of the following would have been helpful:
- Description of the design of the questionnaire the questionnaire is described in the appendix
No. Perhaps there is another appendix which I did not get, but the only thing I see in the appendix is “Contents of the questionnaire”.
Results
What was your response rate? 100%?
Line 202: What did you mean by agree vs completely agree? Level of importance (1-5) perceived, as shown in the questionnaire.
This is not clear. What likert scale did you use and can you list them out? This is part of what could be helpful in the “Description of the design of the questionnaire” I mentioned earlier. Also, what type of questions were they – open, closed ended, or mixed? What were the themes covered? Was it piloted? Why/why not? Was any coding/analysis done for the open-endeds and what type?
Discussion
It would have been helpful to also capture the following:
- Reasons behind those results findings either in the questionnaire or in the discussion.
- Factors influencing doing a physical activity in these women
- Barriers to using the app and how to mitigate
These topics are very interesting, but they were not objectives of this study. Future studies will take these suggestions into consideration. Thank you.
Those were just a few examples to consider. The discussion as is could do with more critical analysis, interpretation of study findings, and comparisons and contrasts to existing literature. I do not imagine that future studies will do so on behalf of this study.
- A pdf or Word document of the actual questionnaire instead of a google doc link that is not in English language. the questionnaire in English is described in the appendix.
Perhaps there is another appendix which I did not get, but I did not see it. What I saw was an outline of the contents in the questionnaire, not an actual sample.
Conclusion
Line 301: Perhaps I missed that information, but I did not see any result finding that confirmed the pregnancy specific app did not exist or that they would use it if it did. Section 3.6 shows that most women use apps. However, most women do not use pregnancy-tailored apps because they do not exist. This is the main reason for conducting this preliminary study.
The reason for conducting this preliminary study was stated. This study did establish that there were no Portuguese pregnancy-specific apps, and that many Portuguese women have not used the app. Also, they agreed that it was important to have the app, and some provided suggestions to the making of an app. However, it is not clear why 10.5% of women have used the app but 89.5% have not. Ninety-eight (98%) agreed it is important to have an app, but it is not clear if they were interested in one or would be willing to use it if it existed. Moreso, only about a quarter were willing to provide any suggestions. It is also not clear if they would prefer to use a pregnancy specific app over the regular app. Factors were not explored to see what could help or not. Thus, it is difficult to agree with the study conclusion that the reason the women do not use the app is because the app does not exist. The proof linking both is not clear.
Comments on the Quality of English LanguageThis is better.
